# Training and Profile of Special Olympics Portugal Coaches: Influence of Formal and Non-Formal Learning

**DOI:** 10.3390/ijerph18126491

**Published:** 2021-06-16

**Authors:** Pedro Pires, Marco Batista, Daniel A. Marinho, Antonio Antúnez, Helena Mesquita, Sergio J. Ibáñez

**Affiliations:** 1Research Group in Optimization of Training and Sports Performance (GOERD), University of Extremadura, 10003 Caceres, Spain; pedroruiinespires@gmail.com (P.P.); sibanez@unex.es (S.J.I.); 2Sport, Health & Exercise Research Unit (SHERU), Instituto Politécnico de Castelo Branco, 6000-266 Castelo Branco, Portugal; marco.batista@ipcb.pt (M.B.); hmesquita@ipcb.pt (H.M.); 3Research Center in Sports Sciences, Health Sciences and Human Development (CIDESD), Universidade da Beira Interior, 6201-001 Covilhã, Portugal; dmarinho@ubi.pt; 4Centro Interdisciplinar de Ciências Sociais da Universidade Nova (CISC.NOVA), 1070-312 Lisboa, Portugal

**Keywords:** learning development, coaches, adapted sport, intellectual disability

## Abstract

Since coaches play an important role in the development of athletes, the process and mechanisms used by Special Olympics Portugal to develop coaches’ skills are worthy of research. In this context, the study aims to identify the training paths and profiles of the Special Olympics Portugal coach. It also aims to analyze the relationship between formal and non-formal learning in the profile and training of this type of coach. The research is descriptive and transversal regarding Special Olympics Portugal coaches, with the participation of 50 subjects. Two questionnaires were used, the Coaches’ Training Profile Questionnaire to determine the training routes, and the Coaches’ Orientation Questionnaire. The results show that the Special Olympics Portugal coaches have an academic background and a somewhat critical profile. It is imperative to build formal and non-formal learning contexts that focus on the theme of adapted sports, in order to allow the training of more qualified coaches, who are consequently more effective in their interventions with this type of athlete.

## 1. Introduction

The professional development of coaches is one of the most important topics in studies focusing on training [1]. However, research on the role of the trainer is weak regarding the interconnection between theory and practice since conceptual procedures in training are still insufficient [2]. There is little research that identifies the profiles of coaches and the variables that condition their professional intervention [3]. In the case of adapted sports (AS), the need becomes more visible, as issues exploring the coach training and training processes in adapted sport are almost non-existent [4]. In Portugal, Special Olympics Portugal is one of the main national organisations for the development of sport for athletes with intellectual disabilities. Since coaches play an important role in the development of athletes, the process and mechanisms used by Special Olympics to develop the skills of the coach are worthy of research [4,5]. Coaches are the directors of the training process, leading the athletes’ learning. This fact is even more relevant with adapted sports athletes who require more help and supervision.

Within this context, Bentzen et al. [6] conducted a scoping review with the purpose of analyzing the scientific literature on adapted sports coaches between 1991 and 2018. These authors concluded that only 28.2% of the studies were of a quantitative nature, thus recommending the use of this investigative design in future research. These authors consider the research on coaches of athletes with intellectual disabilities relevant, due to the specificities of this type of disability.

Research on training programs for coaches of adapted sport has identified that these programs are not adapted to the specific needs of these coaches and athletes, as they are modeled on those designed for normative sport [3,7,8]. The profiles that coaches can adopt are conditioned by the training they undertake (formal and non-formal) [9]. Coaches use different learning pathways to train, which are very heterogeneous and conditioned by the legislation of each country [10]. On the one hand, coaches undergo formal initial training in order to have the professional training to be able to intervene. This initial training is carried out in coaching courses or university training. On the other hand, the continuing education of coaches can be formal (courses, congresses, clinics, continuing education classes, as an assistant to another coach, etc.), or non-formal (communities of practice, peer-to-peer exchange, reading of bibliographies, reflection on actions, observation of other coaches, etc.). Furthermore, the coach’s intervention is conditioned by the coach’s implicit theories, as well as his or her previous experience as a player and coach.

There is a need for research on the profiles of adapted sport coaches, as well as their training, not only because of the lack of research on the subject, but also because of the need to understand the learning contexts of these coaches in order to analyze their formal and non-formal learning paths. A lack of specific learning resources and gaps in the training pathway for adaptive sport coaches with little formal training have been identified. Therefore, coaches resort to acquiring their knowledge through non-formal, non-institutional channels, through peer exchange [8,11,12].

In addition, adapted sport coaches work with a population of athletes with special characteristics that condition their professional intervention. Different stressors that may be experienced by staff working with people with intellectual disabilities have been identified [13]. These factors should be taken into account when analyzing this population of coaches, as they can condition their intervention.

In this context, formal training is an activity that requires admission guidelines, compulsory attendance, and standardized curricula that culminates in certification [14]. Non-formal learning training is a systematic educational activity, carried out outside the formal learning framework [14]. Nelson et al. [14] mentioned that although there are similarities between the two types of learning, non-formal learning training is typically developed in a short period of time on a specific area of knowledge. It should be noted that in the context of AS, research exploring Special Olympics coaches is limited [5], as research focuses more on athletes than on coaches. Coach education is the sum of formal and non-formal learning and experiences as a player and coach, therefore being a complex and holistic process. 

Based on the theoretical construct on coach profiles [15] and the variables that affect the establishment of coach profiles [16], Feu et al. [17,18] propose that the coach’s profile be evaluated from a multidimensional perspective, considering the knowledge and skills used by the coaches themselves [9,10,11,12,13,14,15,16,17,18,19]. These tools have been used in standardized sport to identify the relationships between coach profiles, decision style, planning style and training profiles [20,21], but they have not yet been explored in adapted sport. Burkett [22] states that technical and sports sources that are available on the subject of training and coaching tend to be designed by specialists or academics to meet their own needs, are not accessible to coaches of adapted sports, and are often not suitable to fill gaps in knowledge for the professionals. Identifying the existing relationships between the profiles of adapted sports coaches and the training profiles will provide knowledge about this professional reality.

This study aims to identify the training routes and profile of the Special Olympics Portugal Coach (SOPC). In addition to the above, it also aims to examine the relationship between formal and non-formal learning training in the SOPC’s training and profile, as well as to analyse the correlations between the profile and the training.

## 2. Materials and Methods

The research is descriptive and transversal regarding the Special Olympics Portugal Coaches, with the participation of 50 participants. The data were collected through a questionnaire survey with a convenience sample [23]. All the study procedures were approved by the University’s bioethics board before the study began.

### 2.1. Sample

15 women and 35 men participated, with an average age of 40.22 years and 5.06 years of average experience in coaching adapted sports. According to a source from Special Olympics International, 221 coaches are registered in Portugal. In this context, the margin of error of the study sample is 12.22% within a 95% confidence level.

### 2.2. Instrument

To identify the training routes, we used the Coaches’ Training Profile Questionnaire (QPFT) which is composed of 15 items, divided into five categories, and establishes three sources of theoretical knowledge: Academic Training; Professional Experience and Athlete Experience [16]. The Coaches’ Orientation Questionnaire (COQ) was used to determine the trainer’s profile. This instrument is made up of 46 items corresponding to six coach profiles: Critical; Dialoguing; Technological; Innovative; Traditional; and Collaborative [15,17]. Both instruments are composed of a Likert scale of 11 with fractions of 10 in 10 points (0 = totally in disagreement and 100 = totally in agreement).

### 2.3. Variables

The training actions performed by the coaches were categorized as independent variables. Formal and non-formal training were defined as variables. Different categories were established for each of them (Table 1).

In this context, Gilbert et al. [24] suggest empirically investigating the formal training path and areas of expertise, referring to them as a contribution to the overall process of coach development. The non-formal learning training variable was categorized by the number of actions attended in the last three years, this being the time period established in the Portuguese Republic’s daily newspaper through ordinance 141/2020 (Office No. 141/2020 of the Ministry of Education and Science, Diário da República, 115, June 16, 2020), for the renewal of the coach’s license. This categorization supports the need to strengthen and specify this type of training [16], given that in the adapted sport area these learning moments are very scarce [3]. The dependent variables in the study were defined through the data collection instrument used in the study (Table 2), which were drawn up in accordance with the theoretical framework developed by Ibáñez [15,16,25] and refined by Feu et al. [12,13,14].

### 2.4. Procedure

All study procedures were approved by the University’s bioethics board before the start (ref number 238/2019). The recruitment of participants was carried out using multiple methods. After analysing the timetable of the Special Olympics Portugal (SOP) events, a selection was made of events with a national scope, which included several sports in the programme and lasted more than one day.

In this way, permission was requested from the SOP to distribute the research material in person at the events selected according to the established criteria. At the beginning of the event, all the coaches were contacted personally and if they agreed to participate in the study, they were provided with all the study materials (letter of presentation of the study, informed consent, and survey questionnaire). The coaches had the entire event to choose the time of completion and the researchers were present to clarify any procedure. Before the end of the events, the participating coaches handed in the questionnaires. It should be noted that the face-to-face survey option was carried out before the pandemic context emerged.

To include more coaches, and after the face-to-face process, an online version of the survey by questionnaire was made available. In this way, SOP was asked to collaborate in sending an e-mail to all coaches requesting their participation in the study, as well as all research materials. It should be noted that the coaches who filled in the questionnaire at the events in person were excluded from the online process.

### 2.5. Statistical Analysis

An internal consistency analysis of the QPFT and COQ scales was performed using Cronbach’s alpha coefficient, which obtained an optimum reliability with the value α > 0.70 [26,27]. In the initial phase, the first exploration of the data was carried out using the central tendency measures and the Kolmogorov–Smirnov normality test. The extrapolation of the results was carried out using the data obtained through descriptive statistics. After verifying that the data revealed a normal distribution, the analysis of the variance (ANOVA) was carried out between the profile and the training of the coach with the variables formal training and non-formal learning training. η^2^ was used to analyze the effect of the ANOVA size index, these being classified as: no effect if 0 < η^2^ ≤ 0.04; minimum if 0.04 < η^2^ ≤ 0.25; moderate if 0.25 < η^2^ ≤ 0.64; and strong if η^2^ > 0.64 [28]. 

The degree of correlation between profile and SOPC training was ascertained using Pearson’s correlation coefficient. The relationships were classified as follows: 0 = no correlation, 0 < |r| < 0.2 = very weak correlation, 0.2 ≤ |r| < 0.4 = weak correlation, 0.4 ≤ |r| < 0.6 = moderate correlation, 0.6 ≤ |r| < 0.8 = strong correlation, 0.8 ≤ |r| < 1 = very strong correlation, and 1 = perfect correlation [29]. 

Finally, it is essential to mention that the (chi-square) dependence between formal and non-formal learning training was evaluated using a convergence table, finding that the variables under analysis have a strong association *(p* ≤ 0.001), proving to be good indicators for evaluating the formative paths of SOPCs.

## 3. Results

In the descriptive statistics of QPFT, the SOPCs had higher average values in academic training (*M* = 79.64 ± σ = 12.97), followed by professional experience (*M* = 71.52 ± σ= 19.34) and lastly from athlete experience (*M* = 65.04± σ = 13.56). In the COQ, coaches demonstrated a predominance of critical (*M* = 83.55 ± σ= 8.68), innovative (*M* = 82.98 ± σ = 9.15) and dialoguing (*M* = 80.83 ± σ = 10.36) profiles. The technological (*M* = 76.73 ± σ = 12.18), collaborative (*M* = 76.34 ± σ = 13.39) and traditional (*M* = 74.63 ± σ= 11.98) profiles received the lowest scores. 

Figure 1 describes the average values between the training and profile variables and the specified formal SOPC training.

Figure 2 shows the mean values and standard deviation of the results of the training variable and the profile of the SOPC in non-formal learning activities.

Figure 3 shows the relationships between the SOPC training and the SOPC profile. Initially, a bivariate analysis was carried out in order to determine the relationship between the variables. In this context, we used the Pearson coefficient to analyze the intensity between the training and coach profile variables and to determine the significance of the correlations between them. In the analysis of the results, only positive correlations were revealed, with two types of significance: weak positive and moderate positive [29].

Table 3 shows the analysis of variance (ANOVA) and η^2^ for the analysis carried out by comparing the profile and SOPC training with its formal and non-formal learning training.

Within each group of coaches defined by their training and profile, there are no differences in the formal and non-formal training activities carried out.

## 4. Discussion

The study aimed to identify the training routes and the profile of the Special Olympics Portugal Coach. In addition to the above, it was also intended to examine the relationship between formal and non-formal learning training in the SOPC’s training and profile, as well as to analyze the existing correlations between the profile and the training.

The results show that SOPCs tend to have academic experience, followed by professional and athlete experience. The coaches have mixed profiles, where the critical, innovative, and dialoguing profiles predominate.

These results are corroborated in the research by Feu et al. [17], who state that the more knowledge and academic influence coaches have, the less confidence they have in their training due to their athlete experience. However, in adapted sport, coaches with disabilities are scarce, so few of them have experience as players in the sport [30,31]. The research mentioned above considers that experience as a player in the field of disability increases their knowledge, experience, and effectiveness as coaches of adapted sport. The critical profile has the highest average value. Research also indicates that this is a predictable outcome as there are few learning contexts in the area of adapted sport [12].

In formal training, no statistically significant differences in SOPCs are identified. Despite this, there is a tendency for adapted sport coaches to define themselves as more dialogic and less technological. In addition, coaches with a Physical Education and Sport Sciences background are identified as having a more critical and less traditional profile, while coaches with a technical background are more innovative and less collaborative. These trends provide novel information about the self-defined profile of SOPCs. Regarding training, the coaches have higher average values in academic training and less in training as athlete experience. In the study carried out by Fraile et al. [32] who analyzed the profile of school sports coaches from Portugal, France, Italy and Spain, the results found are in agreement with those of our research. It is identified that there are no pure coach profiles. As in adapted sports, coaches define themselves and they consider that it is better for their professional intervention to adopt more dialogic profiles, with reflective attitudes and willingness to innovate in the training process. Training with young athletes and adaptive athletes may require a greater mastery of these skills than others. Thus, the results of this research confirm that the profiles with the greatest prevalence are dialoguing, followed by critical and innovative profiles. On the other hand, the collaborative, traditional and technological profiles are the ones with the lowest average scores. The characteristics and needs of adapted athletes may require less mastery of technological skills, the adoption of a traditional profile or the delegation of skills, as training is highly personalized. In other research, mixed profiles are identified [17]; in addition, trainers define themselves on more occasions as dialoguers, critics, and innovators, and on fewer occasions as collaborators, traditional and technological.

Regarding non-formal learning, all coaches, regardless of their profile and training, have carried out at least one training activity in the last 3 years, except for the athlete experience coach profile, who, on average, have carried out more non-formal training activities. Particularly in AS, coaches carry out non-formal learning actions due to the lack of formal education actions specific to adapted sports [7,8,9,10,11,33]. Indeed, Fairhurst et al. [12] find that training opportunities in the context of AS are almost non-existent compared to regular sport. This premise leads coaches to look for alternative sources of knowledge, which consequently provide learning that tends to be acquired through trial and error [12].

Regarding the correlations between the profile and the SOPC training, we find that they are all a positive typology. It can be seen that the dialoguing profile relates to all other profiles and training, with the exception of training as an athlete experience. In addition, it should be noted that the critical profile has correlations with four dimensions of the profile and one of the training, and the innovative profile has four exclusive correlations with the dimensions of the profile. The coaches have mixed profiles [34,35], facilitating the assistance and understanding of the specific needs of athletes with disabilities [8]. Academic training allows the adoption of specific coach profiles, such as traditional, dialogic, and critical. The knowledge acquired in academic institutions facilitates reflection and dialogue between the coach and the athletes. This capacity for dialogue is also possessed by professional experience coaches. In training with children, coaches with academic training are more effective as they can use a greater range of resources and skills [36]. 

The results have highlighted the importance of academic training to adopt a specific set of coach profiles working with adapted athletes. From this perspective, it is necessary for researchers, educators, and coaches to critically examine the assumptions about the training of disabled athletes and the implications for coach learning, education, and practice [3]. It is necessary to analyze the training programs with which these coaches are trained, allowing them to develop the specific competencies demanded by these athletes. Thus, for coaches who work with disabled athletes, knowledge and know-how become essential, as they have the dual function of understanding sport and athlete typology [7]. There is a clear need to improve the training of SOPCs, especially in the development of learning situations [8,11]. However, there is also a need to place greater demands and challenges in the guidelines for the training of coaches of adapted sport, in order to understand and establish new suggestions for future training in this area [12]. 

The stress endured by the coach of disabled athletes is a factor that conditions their professional intervention [13]. Coaches should be offered activities that allow them to be trained to better cope with these stressful situations, both within formal and non-formal training.

From this perspective, it is important to understand the reality of the SOPCs in greater depth and the overall context by including non-formal learning and reflection in future studies. In addition, it would be interesting to carry out the same study with other types of disability, as well as to analyze the decision and planning styles of the coaches. Furthermore, in future studies with coaches of adapted sports, the stress variable should be included as a contrast to their intervention. 

## 5. Conclusions

The results reveal that the SOPCs have mixed profiles, where the critical, innovative, and dialoguing profiles predominate. This situation implies that the coaches in this area are more reflective, needing the discourse to work with the athletes and look for new solutions in order to adapt the training to the effective needs of the athletes.

In the training parameter, all coaches use academic training as the most recurrent source. This can be explained by the fact that most coaches have formal training with the completion of a higher education. However, few coaches have specific knowledge in the area of disability. It is therefore necessary to reinforce the need for both formal and non-formal learning covering both areas. It should be noted that there should be a concern to train the players in adapted sport, since there are few coaches with disabilities. This premise is essential, as experienced players are proving to be an important source of learning, which is being wasted due to the lack of formative importance accorded it. 

This study confirms that coaches seek non-formal learning contexts, reinforcing their need to obtain information for the development of their skills. In short, it is important to highlight the urgent need of constructing formal and non-formal learning contexts that focus on the theme of adapted sports. This would allow a more qualified training of coaches, and consequently the presence of more effective coaches in their interventions with this type of athlete.

## Figures and Tables

**Figure 1 ijerph-18-06491-f001:**
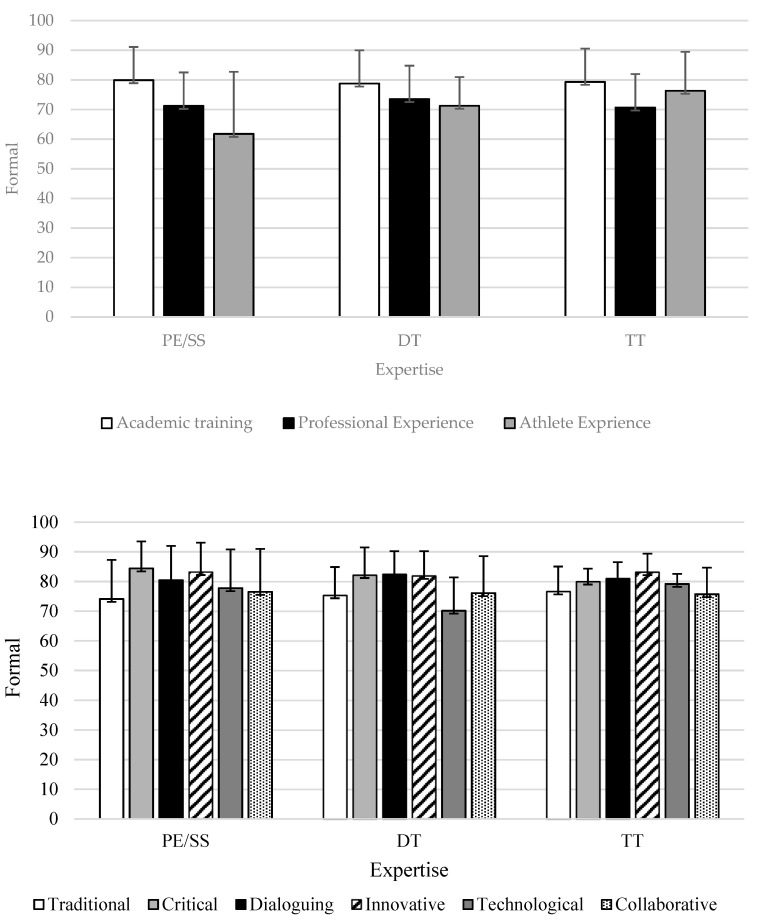
Descriptive statistics of the relationship between the training and profile of SOPC and formal learning.

**Figure 2 ijerph-18-06491-f002:**
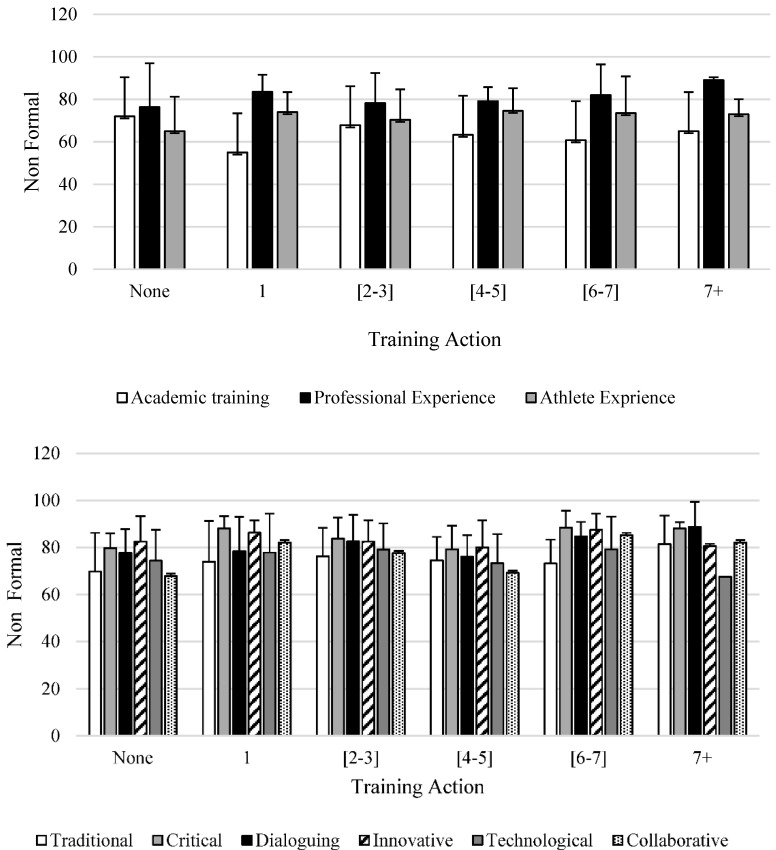
Descriptive statistics of the relationship between training and profile of SOPC and non-formal learning training.

**Figure 3 ijerph-18-06491-f003:**
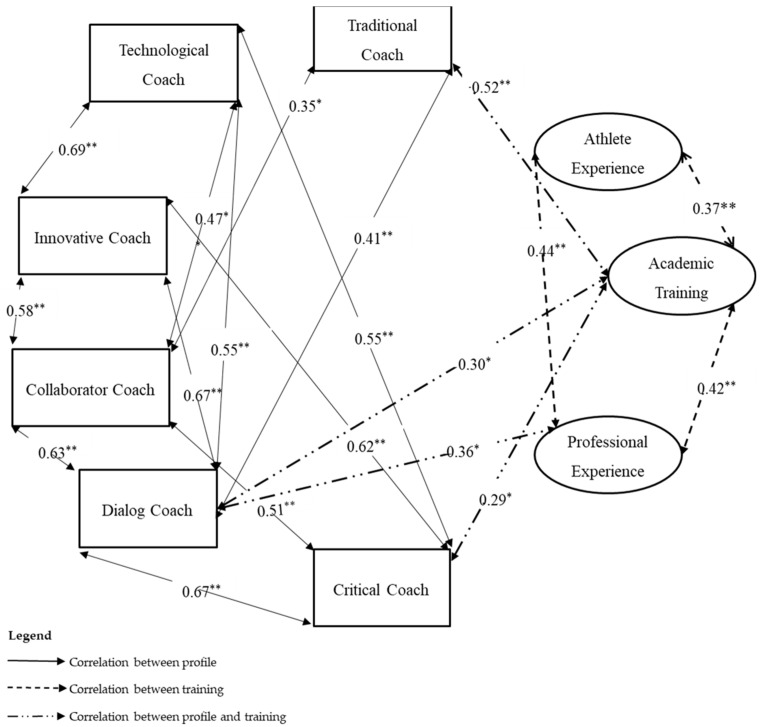
Correlations between the profile variables and SOPC training. * *p* < 0.005; ** *p* < 0.001.

**Table 1 ijerph-18-06491-t001:** Categorization of formal and non-formal learning training variables.

Variables	Categorisation	N	%
Formal Learning	Physical Education and Sport Sciences (PE/SS)	36	72
	Disability Training (DT)	8	16
	Technical Training (TT)	6	12
Non-formal Learning	1 training action	3	6
	2 and 3 training actions	21	42
	4 and 5 training actions	10	20
	6 and 7 training actions	8	16
	+7 training actions	2	4
	None	6	12

**Table 2 ijerph-18-06491-t002:** Definitions of the trainer models: Training and Profile (Adapted from Ibáñez [15,16]).

	Dimension	Definition
Training	Academic Training	Coaches whose knowledge is acquired in training institutions, and their experience in the sports field comes fundamentally from the study of training.
	Professional Experience	Coaches who train themselves by researching, innovating, and applying theories formulated by themselves.
	Athlete Experience	Coaches who have been players, i.e., a recycled player. They tend to reproduce the models and attitudes they have experienced as players, selecting the most striking ones.
Profile	Traditional Coach	Coaches who transmit, as a priority, models of recognised effectiveness, through a directive style of teaching. They prefer a serious and tense training atmosphere where the players know what they must do, and their assistants follow their instructions.
	Technological Coach	Coaches who base their actions on the study and control of the factors that influence their sport. They need their technical assistants to be experts in measuring and analysing these factors and their players must be willing to cooperate.
	Innovative Coach	Coaches characterised by innovative training strategies and elements. They experiment and introduce changes to seek greater effectiveness and prefer their technical assistants to suggest innovations to improve training. Players are sometimes confused by so many changes.
	Collaborative Coach	Coaches who prefer to delegate functions to specialist employees in different facets, because it is difficult for them to be an expert in all facets of training. They maintain a climate of trust with the players and the assistants, who are often the intermediaries between the head coach and the player.
	Dialoguing Coach	Coaches who try to control through dialogue all the elements surrounding the training, media, management, technical assistants, and players to convince them of the work being done, thus promoting a good training climate.
	Critical Coach	Coaches who analyse, reflect on, and criticise the training process they are developing and are therefore not conformists. This premise leads them to create a tense climate in their work.

**Table 3 ijerph-18-06491-t003:** Results of the analysis of the differences between training and SOPC profiles.

		Formal Learning			Non-Formal Learning		
		*F*	*p*	η^2^	*F*	*p*	η^2^
Training	Athlete Experience	2.03	0.14	0.29	0.52	0.76	0.15
	Academic Training	0.03	0.97	0.11	0.43	0.82	0.13
	Professional Experience	0.10	0.90	0.19	0.46	0.81	0.14
Profile	Traditional Coach	0.13	0.88	0.07	0.40	0.85	0.12
	Critical Coach	0.46	0.46	0.14	1.15	0.15	0.22
	Dialoguing Coach	0.88	0.88	0.19	0.33	0.33	0.12
	Innovative Coach	0.94	0.94	0.20	0.75	0.59	0.18
	Technological Coach	0.25	0.25	0.10	0.65	0.67	0.17
	Collaborative Coach	0.99	0.99	0.20	2.34	0.06	0.31

## Data Availability

Not applicable.

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
