# Peer review of "Training and Profile of Special Olympics Portugal Coaches: Influence of Formal and Non-Formal Learning"

_ijerph, 2021, doi:10.3390/ijerph18126491_

Round 1

Reviewer 1 Report

The following list are suggested revisions to consider when revising the current manuscript presented according to line number.

Line 29. Replace ‘subject’ with the term topic in the sentence, “The professional development of coaches is one of the most important subjects topics in studies focusing on training”. In addition, it is unclear what is meant by training? The context needs to be more clearly stated.

Lines 29-31. Unfortunately, this statement is confusing and several aspects are not clear – for example, what are the authors meaning by ‘conceptual procedures in training are still insufficient’?

Lines 31-33. Given the different contexts that are possible for the term 'training', it becomes difficult to know what type of training is being discussed at what time. For example, in lines 31-33, the term training is used twice (i.e., once in regards to the training of coaches (professional development of coaches and/or coaching education); while the second type of training is less clear. To confirm, is the latter referring to the training programs of the athlete?).

Lines 28-37. Is the 'trainer' being used interchangeably with the term coach? If yes, this is likely leading to confusion when reading. Usually a trainer (e.g., qualified exercise physiologist, athletic therapist, strength and conditioning trainer) is different than the coach. Each has different roles in a multidisciplinary team. This meaning of trainer needs to be clarified.

Line 38. It is unclear what, ‘faced with this paradigm’ means? The review by Rangeon et al.  identified the methods and approaches to the field in this area – was it not? Was the review not establishing the paradigm?

Line 38. Replace ‘subject’ with the term, topic.

Line 39. It is unclear in the statement, “only 4 explored the problem of training in AS” why it is written as a deficit-based statement and why training is identified as a problem? Are these studies examining coaching development in Adapted Sports?

Line 40. What is meant by a general development regarding Adapted Sport?

Lines 40-41. The following sentence is not clear: “…which has not been effectively accompanied by the production of scientific research, particularly in areas such as training, the coach and training programmes [6].” What is meant by the coach? What is the difference between training and training programmes?

Lines 43-45. This is a one sentence paragraph! Revise.

Line 46. The use of ‘acquires’ is grammatically incorrect in the sentence “This study acquires relevance…”. Revise.

Line 46-47. There has been no rationale or background information provided for a research focus on ‘formal and non-formal learning paths’.

Line 50. Need to provide a rationale and/or explanation as to why acquiring knowledge through peer exchange is undesirable. The authors are implying that none of this knowledge can be beneficial.

Line 54. Who are the authors? Are you referring to Reference #11?

Line 58. Is it unclear what the ‘the above-mentioned problem’ is?

Lines 58-59. Please define what is meant by coach’s profile and how this is different from a coaches background experience or developmental pathway? These differences in terms/concepts are not well differentiated and/or a rationale is lacking as to why the different constructs should be investigated at the same time.

Line 70-71. Remove: The Materials and Methods should be described with sufficient details to allow others to replicate and build on the published results.

Line 73. Please remove the term ‘subjects’ and replace with ‘participants’. People provide informed consent to participate in research, they are not subjects.

Line 77. Remove ] in the sentence, “The study involved 50 SOPCs)…”.

Lines 82-85. A clearer rationale needs to be developed in the introduction to the manuscript around such aspects as coaching orientation in relation to the purpose of the study. Currently, the implementation for the two questionnaires (Coaches' Training Profile Questionnaire (QPFT) and The Coaches' Orientation Questionnaire (COQ)) needs to better relate to the text of the introduction. For example, how does looking at the various orientations of a coach (e.g., traditional) connect to formal and non-formal training. Synthesize and present the literature according.

Lines 91-92. The variables identified for formal and non-formal are not consistent with the questionnaires administered. How can a coach be categorized as formal training or nonformal training? Training type is an environment one is exposed to; it is not a quality of a person. Are you categorizing coaches backgrounds and/or experiences? Overall, the discussion on variables is very difficult to follow.

Table 2. This table is very difficult to read. Perhaps widen the Dimension Column, to increase the ease of reading the Table.

Line 109. This line is repetitive. It is already stated in lines 74-75.

Line 188-189. Revise “..followed by professional and lastly athlete experience” to “…followed by professional and athlete experience, respectively”.

Line 192. Remove ‘the’ before coaches from the statement, “…the more knowledge and academic influence the coaches have…”.

Line 193-194. Repetitive use of the word ‘few’, revise.

Lines 199-203. There are no statistically significant differences for formal training as identified by the authors, yet sentences 199-203 are written as if significant differences exist.

Lines 203-204. What do the authors mean by ‘a recycled player’.

Lines 206-209. The authors refer to profiles with the greatest relevance; but what does this mean? Relevance to what? What does it mean for the practical setting if one scores low?

Lines 210-213. The following sentence is unclear: “When it comes to non-formal learning, both in profile and in training, all the coaches obtain as a higher average value the frequency of at least one training action in the last 3 years, with the exception of the training of a retrained coach where there is a higher average value in the frequency of non-formal training actions. Further, what is a ‘retrained’ coach?

Lines 213-214. What does the term effectively mean in this context? What is the behaviour – finding non-formal learning contexts?

Line 217-218. Why is trial and error learning identified as an undesirable process? There is an entire field of literature in motor behaviour (i.e., motor learning literature) that would demonstrate the importance of trial and error for learning. There is no rationale and/or evidence presented in the manuscript that would explain why this is not an effective learning strategy.

Line 221. Again, what is a recycled player?

Lines 219-227. Here is an example of results being re-presented. There should be greater discussion around what the results mean.

Line 229. It is stated that, “there is an urgent need for researchers, educators, and coaches to critically examine the assumptions about the training of disabled athletes”. This idea is a departure from what has been presented/discussed in the manuscript up to that point. What are the postulated assumptions?

Line 239. In future research it is suggested to study other types of disability; however, it is unclear what specific disabilities were focused on in the present study.

Line 255-256. One sentence paragraph!

Author Response

Dear Reviewer,

We have carefully considered all considerations in the document provided by you. Enclosed you will find our detailed answers to your inquiries.

Reviewer 2 Report

The authors have done a nice study filling an important gap regarding the training and profile reporting of coaches for individuals with intellectual disability. There are some minor concerns which I believe when addressed could make this study stronger.  

Overall comments: It is not very clear that your study focused on coaches working with special populations with intellectual disability. Please make that clear otherwise the reader could get the message that your study could be a mix of intellectual and physical disability. May be the title of your study could be specific and include intellectual disability words. Also, when you talk about coaches having experience informing coaching, you need to make sure that you mean coaches with ID. I hope that makes sense.

Also, please add limitations of your work.

Specific comments:

Line 32, page 1: It is not clear when you say: training of the coach and training. Are these different or same? Pl. clarify.

Line 35, page 1: Please clarify how coaches play a critical role in the development of special athletes. Adding this info would make your argument of examining skills of trainer for special Olympics stronger.

Line 42: same comment as line 32, page 1 comment.

Line 39, page 1, what do you mean by the problem of training? Pl. clarify.

Line 46, page 2: You have not yet said what your study is about so he start of this sentence needs to be modified.

Line 57, page 2: what do you mean by ‘the subject of special Olympics coaches’? it is a vague term. Please be specific.

Lines 70-71, page 2: Is this a typo?

Line 78: pl provide the mean years of coaching experience.

Line 79: margin of error of what? Pl clarify.

Line 91: pl provide either’:’ or ‘of’ after the words independent variables

Table 2: In your definition of are the following lines examples of subjective language or your interpretation or are they part of the definition of that coaching? If the following sentences are inferential and not constitutionally define that specific type of coaching then it is good o get rid of those lines but if the following lines constitute the definition then those can remain. Hope I am clear.

Following are the lines which could be considered subjective:

Dialoguing coach: ‘thus promoting a good training climate’.

Innovative coach: Players are sometimes confused by so many changes.

Traditional coach: ‘They……instructions’.

Critical coach: ‘This premise…….work’.

Line 111: Pl mention SOP before starting to use it in later lines.

Iine 184: Pl use SOPC before starting to use it for the 1st time  in line 186.

Line  204: It is intriguing that school sports coaches profile matches yours. Why could that be? Was it surprising or did you expect that? Pl explain.

Line 208: Why do you think ‘the collaborative, traditional and technological profiles are the ones with the lowest average scores’? Pl explain.

Under conclusion: your 2nd and 3rd para could be combined to tell the full picture together.

Author Response

(The authors gave the same response as above.)

Reviewer 3 Report

The manuscript presented for evaluation is methodologically correct. Research on trainers of disabled people is interesting and rare in the literature. The authors adopted two test tools: QPFT  and COQ. These tests have a large amount of literature. However, the coaches of athletes without disabilities were assessed. Unfortunately, neither in the introduction nor in discussion the important element of the trainer's work was noticed. It is a stress that is different in disabled athletes than in non-diseased athletes. A study by Hatton (https://doi.org/10.1016/S0891-4222(99)00009-8Get rights and content) carried out in Great Britain shows that 1/3 of staff working with people with intellectual disabilities experienced high levels of occupational stress. They are accompanied by worrying mental health indicators It is not understandable why such an important element of trainers' evaluation was omitted by the authors. In the opinion of the reviewer, this manuscript cannot be considered complete without justification in the Introduction section and extensive discussion supported by the literature. The research results obtained are inconsistent in many places. Perhaps these inconsistencies can be explained by the high level of stress associated with working with people with disabilities. I expect supplementation of the presented work and justification for omitting the stress assessment, eg Staff Stressor Questionnaire (SSQ) in the surveyed trainers. Then you can start a comprehensive job evaluation.

Author Response

(The authors gave the same response as above.)

Round 2

Reviewer 3 Report

He maintains his reservations about the research project. Additions and corrections allow for publication